# Application of Magnetite-Nanoparticles and Microbial Fuel Cell on Anaerobic Digestion: Influence of External Resistance

**DOI:** 10.3390/microorganisms11030643

**Published:** 2023-03-02

**Authors:** Nhlanganiso Ivan Madondo, Sudesh Rathilal, Babatunde Femi Bakare, Emmanuel Kweinor Tetteh

**Affiliations:** 1Green Engineering Research Group, Department of Chemical Engineering, Faculty of Engineering and the Built Environment, Steve Biko Campus, Durban University of Technology, S4 Level 1, Durban 4000, South Africa; 2Department of Chemical Engineering, Faculty of Engineering, Mangosuthu University of Technology, Durban 4026, South Africa

**Keywords:** external resistance, magnetite-nanoparticles, microbial fuel cell (MFC), sewage sludge, anaerobic digestion

## Abstract

In this paper, the application of magnetite-nanoparticles and a microbial fuel cell (MFC) was studied on the anaerobic digestion (AD) of sewage sludge. The experimental set-up included six 1 L biochemical methane potential (BMP) tests with different external resistors: (a) 100 Ω, (b) 300 Ω, (c) 500 Ω, (d) 800 Ω, (e) 1000 Ω, and (f) a control with no external resistor. The BMP tests were carried out using digesters with a working volume of 0.8 L fed with 0.5 L substrate, 0.3 L inoculum, and 0.53 g magnetite-nanoparticles. The results suggested that the ultimate biogas generation reached 692.7 mL/g VS_fed_ in the 500 Ω digester, which was substantially greater than the 102.6 mL/g VS_fed_ of the control. The electrochemical efficiency analysis also demonstrated higher coulombic efficiency (81.2%) and maximum power density (30.17 mW/ m^2^) for the 500 Ω digester. The digester also revealed a higher maximum voltage generation of 0.431 V, which was approximately 12.7 times the 0.034 V of the lowest-performing MFC (100 Ω digester). In terms of contaminants removed, the best-performing digester was the digester with 500 Ω, which reduced contaminants by more than 89% on COD, TS, VS, TSS and color. In terms of cost-benefit analysis, this digester produced the highest annual energy profit (48.22 ZAR/kWh or 3.45 USD/kWh). This infers the application of magnetite-nanoparticles and MFC on the AD of sewage sludge is very promising for biogas production. The digester with an external resistor of 500 Ω showed a high potential for use in bioelectrochemical biogas generation and contaminant removal for sewage sludge.

## 1. Introduction

The overdependence on fossil fuels, high greenhouse gas emissions, and rising fossil fuel prices have led investigators to explore environmentally benign energy sources [1,2,3]. Also, the rising population and urbanization have created massive amounts of waste, resulting in poor waste treatment and management, especially in developing nations [4]. The quest for energy that is renewable to meet the ever-rising energy demand and decrease dependence on fossil fuels is nowadays becoming imperative [5]. There is similarly a continuing search for renewable energy technologies that are both economical and environmentally friendly [6]. Anaerobic digestion is a promising renewable energy technology that converts biomass into biogas and the degradation of complex biodegradable substrates. The produced biogas comprises mostly methane (approximately 60%), a significant renewable energy gas that can be utilized to replace fossil fuels for the generation of electricity [7,8]. Other advantages include, reduced greenhouse gas emissions, recovery of energy in the form of methane, and the ability to accommodate high loading rates and remove pathogens. Anaerobic digestion is one of the most widely utilized and economical bioenergy recovery technologies globally. Although the process has been in use for a very long time, conventional anaerobic digestion currently faces instability issues mainly due to an unbalanced substrate, resulting in the unavailability of electrons among different mechanism phases. This normally brings about low biogas formation and a very long hydraulic retention time [9].

The interspecies electron transfer (IET) between archaea and microbes is the key to the performance of the anaerobic digestion process [10,11]. The use of anaerobic additives such as magnetite-nanoparticles can be used to improve the IET in anaerobic digestion. This has several benefits, namely: higher microbial degradation rates, reduced hydraulic retention time, higher biogas production, and higher toxic contaminant removals.

On the other hand, bioelectrochemical systems are promising hybrid techniques of electrochemistry and biotechnology that can economically produce electrical or chemical power during redox mechanisms catalyzed by bacteria [12]. Bioelectrochemical systems can be classified into microbial electrolysis cell (MEC), microbial fuel cell (MFC), microbial solar cell (MSC), microbial desalination cell (MDC), and microbial electrosynthesis (MES), amongst which the MEC and MFC are the most investigated at present [13]. Similar to other kinds of electrochemical cells (such as batteries), both the MEC and MFC comprise two electrodes (cathode and anode electrodes) that are joined with a conductor to create a closed electric circuit. When the redox potential of the oxidation half-reaction at the anode is lower than that of the reduction half-reaction at the cathode, electric power is produced, as the result of a positive cell potential. If not, a power supply located externally (in other words, electrolysis) is required to drive the redox reactions [14].

Microbial fuel cells are emerging as multipurpose renewable energy systems [15]. This is mainly due to the multidimensional usages of this environmentally friendly system, such as electricity generation, hydrogen production, and wastewater/waste treatment [16,17]. The ability to generate electricity makes the MFC the preferred bioelectrochemical system over the MEC. The ability to produce electricity from biochemical matter with the MFC opens up possibilities for the recovery of renewable energy from agricultural waste, sewage sludge, and industrial effluents. Therefore, the MFC could have the dual benefit of generating sustainable, clean energy and removing waste. The microbial fuel cell is dependent on the electrically active microorganisms, otherwise known as exoelectrogens, for the generation of electricity and treatment of wastewater [18]. However, the power generation of an MFC is even now extremely small for their large-scale usage in the treatment of wastewater. These limitations are generally because of the slow electron transfer kinetics from microorganisms to the surfaces of the electrode at the anode [19]. In fact, scaling-up applications of MFC are restricted by the interaction between bacteria and electrode surface, design concepts, electrochemical constraints and multidisciplinary method of electrochemistry and biotechnology that are environmentally-related.

The efficiency of the anaerobic digestion method can be improved by enhancing the interspecies electron transfer. Normally, the potential of the anode, which directly regulates the anode availability as electron acceptors, is controlled by an external resistor. Consequently, the growing competition between the non-electrogenic and electrogenic bacterial populations is affected when different external resistors are employed in the bioelectrochemical systems. External resistance is one of the highly significant factors for the commercialization of microbial fuel cells [20,21]. The performance of MFC is affected by external resistance, as it limits the flow of electrons from the anode electrode to the cathode electrode [22]. Even though the power-generating microorganisms can overcome the system’s overall resistance, choosing the right external resistance can reduce system losses and optimize efficiency [23]. The presence of an external resistance or load resistor is essential in the MFC as it closes the circuit, allowing electrons to flow from the anode to the cathode [24]. When the external resistance is at its optimum level, both the coulombic efficiency and output power increase, whereas the production of methane decreases [25]. Parameters of interest in electronic circuits, such as cell voltage and current rate can be controlled using external resistance [26]. In anaerobic digestion, the use of conductive additives that contain iron, for example magnetite-nanoparticles can be useful in enhancing the IET between hydrogenotrophic and volatile fatty acid (VFA) microbes. Most enzymes in anaerobic digestion are metalloenzymes, which means that they need the presence of metals as co-enzymes. In fact, nearly all metalloenzymes that are in the pathway of biogas generation have several agglomerates containing iron; iron is absolutely required for methane generation and cytochromes. Our previous study [27] investigated the effect of adding 1 g of magnetite-nanoparticles on a microbial electrolysis cell (MEC). The results showed that, the addition of magnetite-nanoparticles improved biogas production (15 mL) and methane content (84%) by up to 3.1 and 1.1 times the 4.8 mL and 79.1% of the MFC, respectively. With the synergy of anaerobic additives such as magnetite-nanoparticles and a biochemical system such as microbial fuel cell proving to enhance the performance of anaerobic digestion (i.e., higher IET, methane yield, electrical conductivity, and contaminants removal rate) [27], no study has ever been made on the influence of external resistance on the synergy of magnetite-nanoparticles and a microbial fuel cell.

Thus, the influence of external resistance on the application of magnetite nanoparticles and microbial fuel cells was extensively studied using MFC digesters with 100 Ω, 300 Ω, 500 Ω, 800 Ω and 1000 Ω external resistances. The main focus of the study was on cumulative biogas yield, electrochemical efficiencies (coulombic efficiency, electrical conductivity, magnetic field strength, maximum current density and voltage generation), decontamination of the wastewater and stability indicators.

## 2. Materials and Methods

### 2.1. Equipment Set-Up and Operation

The experimental work was carried out by comparing five MFCs with different external resistors and a control. According to Li and Chen [14], the optimum external resistance of a bioelectrochemical system such as a MFC lies between 100 and 1000 Ω. Therefore, the experiments of the MFCs were carried out at 100 Ω, 300 Ω, 500 Ω, 800 Ω and 1000 Ω. Figure 1 shows a schematic diagram of the digesters that were used in this investigation. Figure 1a represents the control digester which had no external resistor; the control was fed with 300 mL sewage sludge, 500 mL waste-activated sludge, and 0.53 g magnetite-nanoparticles [28]. Each of the MFCs was fed with the same substrate. In addition, each MFC had zinc (anode) and copper (cathode) electrodes that were connected employing an external resistor as shown in Figure 1b. Each digester consisted of an airtight cap located at the top of the digester with four holes for transferring gas to the biogas collector using a 0.6 cm (inside diameter) tubing connector, sampling, anodic electrode, and cathodic electrode. The zinc electrode (anode) and copper electrode (cathode) were inserted inside the BMPs and had a width of 1 cm as well as a length of 12 cm. The experimental work was carried out for a period of 30 days at a constant mesophilic temperature of 32.2 °C by using a water bath [28].

### 2.2. Analyses and Substrates

Daily biogas production was measured using the biogas collector via the water displacement techniques as depicted in Figure 1 [29]. The water displacement method was used for measuring biogas since it is the cheapest method. This technique can provide biogas measurements within the 5% accuracy [30]. Biogas comprised mostly of 60% methane and 40% carbon dioxide [7]. Thus, only methane and carbon dioxide were considered when compensating for gas dissolved in water. Methane is less soluble in water (i.e., 0.017 g methane per kg of water at 1 atm and 32.2 °C), while carbon dioxide is highly soluble in water (i.e., 1.20 g carbon dioxide per kg of water at 1 atm and 32.2 °C). Nonetheless, the amount of gas that was dissolved in the contents of the water displacement system was taken into consideration using the solubility values from Perry et al. [31]. The water quality parameters measured are chemical oxygen demand (COD), total suspended solids (TSS), volatile solids (VS), color, and total solids (TS). A Hach DR 3900 spectrometer (Hach, Loveland, CO, USA) was used to measure COD, TSS, and color [32]. The TS and VS were analyzed according to water and wastewater methods [33]. Standard methods as proposed by APHA [34] were used to characterize wastewater. For TS, 50 mL of samples were kept at 105 °C by means of a drying oven for 24 h. The dried samples were then inserted in a furnace at 550 °C for 2 h in order to determine VS. Electrical conductivity, pH, and total dissolved solids (TDS) were measured once every 5 days using a Hanna H198129 conductivity meter. After analyzing for electrical conductivity, pH, and TDS, the sample was returned into the digester. A digital multimeter (FLUKE 177 RMS, Everett, WA, USA) was used to measure current, resistance, and voltage daily. A digital Telsameter was used to obtain the magnetic field in the digesters. The physical and chemical properties of the feed are shown in Table 1.

The percentage of contaminants removed coulombic efficiency, current density, and power density were calculated as previously described [35].

Both sewage sludge and waste-activated sludge were attained from a local-based treatment works, in the KwaZulu-Natal province of South Africa. The samples were obtained using 20 L containers.

### 2.3. Synthesis of Magnetite-Nanoparticles and Chemical Reagents

The magnetite-nanoparticles used in this work were acquired from the magnetite-nanoparticles synthesized by Amo-Duodu et al. [36]. Their work involved the analysis of magnetite-nanoparticles physiochemical and morphological characteristics, which were determined via scanning electron microscopy/energy-dispersive X-ray (SEM/EDX), Fourier-transform infrared spectroscopy (FTIR), and X-ray diffraction (XRD). The synthesis of magnetite nanoparticles was obtained via a co-precipitation method. The co-precipitation method included the addition of chemical reagents, namely nickel (II) nitrate hexahydrate, ferrous sulfate heptahydrate, oleic acid, sodium hydroxide, and ferric chloride hexahydrate. Ferric chloride hexahydrate was obtained from United Scientific SA cc, Durban, KwaZulu-Natal, South Africa. Both ferrous sulfate heptahydrate and sodium hydroxide were obtained from Labcare Supplies (PTY) LTD. On the other hand, oleic acid and nickel (II) nitrate hexahydrate were acquired from Sigma–Aldrich, Durban, KwaZulu-Natal, South Africa. The X-ray diffraction verified that the structure of magnetite-nanoparticles had a face-centered cubic shape and the size of the crystal was 5.179 nm.

## 3. Results and Discussions

### 3.1. Biogas Accumulation

The biogas accumulation graph is significant in anaerobic digestion since it displays the microbial growth rate. Figure 2 portrays the cumulative biogas production as a function of time for the digesters: 100 Ω, 300 Ω, 500 Ω, 800 Ω, 1000 Ω and control. The results were normalized according to standard temperature and pressure (STP). The standard conditions were as follows: volume (22.4 m^3^), temperature (273.15 K), pressure (1 atm) and mole (1 kmol). The temperature of the system was 32.2 °C, which is 313.15 K. The pressure of the digesters was assumed to be 1 atm. External resistance is one of the most common factors influencing MFC, which directly affects the start-up time [37]. Increasing resistance normally decreases start-up time or lag phase. Thus, the growth of bacteria takes place faster at very high resistances [38,39]. The same observation was found in our study; during the lag phase (day 1), an increase in external resistance was accompanied by an increase in biogas accumulation, with the 1000 Ω showing the highest production of 51.3 mL/g VS_fed_. After passing the startup time, the digesters approached the exponential stage, and the generation of biogas increased extensively due to the exponential growth of methanogens, with the moderate external resistance of 500 Ω revealing the greatest maximum growth rate. On the other hand, the high resistance of 1000 Ω resulted in the lowest growth rate. After passing through the exponential phase, all digesters approached the stabilized or asymptotic stage. The digester with 500 Ω external resistance stabilized at the highest biogas accumulation of 692.7 mL/g VS_fed_, which was about 6.8 times the 102.6 mL /g VS_fed_ of the control.

### 3.2. Current, Resistance and Voltage

As indicated above, the graph of biogas accumulation over time is very helpful in investigating bacterial growth rates. However, it is also important to investigate the behavior of electrons externally. This can be achieved by exploring parameters that affect Ohm’s law, namely current or current density, resistance and voltage.

Current density is perhaps one of the most important electrochemical parameters in the design of electromagnetic fields. External resistance is a key factor that regulates the generation of current from a MFC when employed as a source of power [40]. Figure 3 shows the variation of current density during the biofilm growth period. In the early stages of the digestion process, i.e., hydrolysis stage, external resistance improved the performance of the MFCs; i.e., higher resistance was accompanied by the high current generation. From this, it can be said that Ohm’s law, which states that resistance is inversely proportional to current, is not valid in the hydrolysis stage. In the hydrolysis stage, the 1000 Ω digester generates the highest current density of 19.1 mA/m^2^ on day 1. After passing the hydrolysis stage, higher external resistance hindered the performance of the MFCs. The current density declined concerning the increasing external resistance as stated by Ohm’s law. The digester with the highest external resistance of 1000 Ω did not stabilize but the current dropped quite significantly. The higher the current density and ohmic resistance, the higher the ohmic losses that the system will experience. Therefore, the resistance of the 1000 Ω digester was probably too high in such a way that it limited the amount of available current to be delivered. When operating at higher external resistance that nearly imitates an open circuit, it was observed that the transport of electrons to an electron receiver did not favor the microbes. In such cases, the higher external resistance limits the current flow in the system, which affected the microbe’s ability to colonize the anodic electrode [41,42]. This collaborates with the investigation by Ghangrekar and Shinde [43], which found that when external resistance is extremely high, the current in the system was brought to minimum. Cai et al. [44] investigated the influence of external resistance and biofilm porosity on the transfer of electrons from redox mediator to the anodic electrode, microbial growth and generation of electricity in MFC. The investigators found that lower external resistance was favourable for the growth of exoelectrogens. The greater exoelectrogens concentration contributed to transferring more electrons to the anodic electrode; exoelectrogens improved the redox reactions at the surface of an electrode. On the other hand, very high external resistance restricted the current, which did not help exoelectrogens to colonize the anodic electrode. The methanogens had diverse population in the biofilm, where suffered fierce competition of substrate consumption with exoelectrogens. When lower external resistance was applied, the exoelectrogens were more competitive compared to methanogens. Therefore, it is likely that the digester with 1000 Ω had more methanogens than exoelectrogens, which resulted in lower current density. Aelterman et al. [45] also came to the same conclusion, when they found that high external resistance leads to lower current.

Contrary to the digester with an external resistance of 1000 Ω, the other digesters stabilized. The best performing digester was the 500 Ω, which stabilized for a longer duration of 10 days (from day 5 to day 14) and at the highest current density of 70.0 mA/m^2^. This suggested that the microorganisms of a 500 Ω digester were well adapted at a higher current than the other digesters.

The effect of the digester type on the overall resistance is displayed in Figure 4. External resistance affected the stability of the digesters; the results showed that the higher the resistance the lower the stability, with the stability better between 100 Ω and 500 Ω. However, for higher external resistance (i.e., above 500 Ω), the digesters showed poor instability. The digester with 1000 Ω external resistance revealed the highest instability; i.e., the difference between maximum overall resistance and minimum overall resistance for the 1000 Ω external resistance was 164 Ω, whereas the difference was 23 Ω for the 100 Ω resistor.

It is evident from the graph that, an increase in the overall resistance was observed when time increased from day 1 to day 30. The increase in overall resistance for digesters with external resistance less than 500 Ω was very small. On the other hand, digesters above 500 Ω showed a higher increase in overall resistance, with the 1000 Ω digester showing the greatest increase. The higher increase in overall resistance of the 1000 Ω digester could indicate the formation of higher biofilm growth at the surface of the electrode. Usually, microbial populations acclimatized to a greater external resistance, as was the case with the 1000 Ω resistor, can decrease the onset anode potential in a MFC [39]. However, this conclusion has to be verified in the voltage and electrochemical efficiency sections that will be discussed later in this paper.

External resistance (R_ext_) influences the efficiency of a microbial fuel cell, as it limits electron flow from the anode electrode to the cathode electrode [46]. As stated by Ohm’s Law (i.e., $V=I\times R_{ext}$), the current (I) will be affected, and consequently voltage (V). The DC voltage of the MFCs was recorded against time for a period of 30 days as displayed in Figure 5. It is evident from the graph that, all MFC digesters generated voltage. For all digesters except for the 1000 Ω digester, voltage generation firstly increased, then stabilized, and finally decreased. In fact, for very low external resistance (i.e., 100 Ω) and very high external resistance (i.e., 1000 Ω), the voltage generated by the digesters turned out to be very small. When external resistance was low, the very small redox potential at the anode electrode possibly made it unfavorable as a receiver of electrons for the microorganisms; when resistance was high, the overall resistance was perhaps too much (nearly imitates an open circuit) for the microorganisms to transport the electrons.

The greatest voltage generation of 0.431 V was obtained from the digester with 500 Ω external resistance, and this was almost 12.7 times the 0.034 V of the least performing digester, i.e., digester with 100 Ω external resistance. At external resistance of about 500 Ω, more positive anode potential assists the microorganisms to acquire more energy, therefore more electricity is generated [21,41].

### 3.3. Electrochemical Efficiencies

Although current, resistance and voltage are useful parameters in electronic circuits, as mentioned above, electrochemical efficiencies offer even better in-depth analyses of the behavior of bioelectrochemical systems in anaerobic digestion. The electrochemical efficiencies of the MFC digesters were represented by the magnetic field strength, power density, coulombic efficiency and electrical conductivity.

In anaerobic digestion, magnetic fields can drastically affect the activities of methanogens and their microbial population shift. The relationship between magnetic field strength and external resistance was investigated using 100 Ω, 300 Ω, 500 Ω, 800 Ω and 1000 Ω resistors and the results were graphically presented in Figure 6. Unlike current density and voltage, the magnetic field behaved differently. Before day 5, external resistance enhanced magnetic field strength. The digesters then reached maximum magnetic field strength after day 5. The increasing order of maximum magnetic field strength depicted the following: 1000 Ω: 7.12 mT >800 Ω: 6.51 mT >500 Ω: 5.21 mT >300 Ω: 3.36 mT >100 Ω: 2.57 mT >0 Ω (control): 0.3 mT. Therefore, high external resistance enhanced the performance of the system in the early stages of the digestion process. The digesters with 800 Ω and 1000 Ω resistances revealed maximum magnetic field strengths of over 6.51 mT. Our previous study [47] showed that above 6.24 mT, the bioelectrochemical system is inhibited. Therefore, the higher magnetic field strengths (>6.24 mT) in both the 800 Ω and 1000 Ω hindered the system, resulting in a drastic fall in magnetic field strength after day 5. Consequently, in the end, the 1000 Ω resulted in the lowest magnetic field strength amongst the MFCs.

As was the case with resistance, magnetic field strength also affected stability; the results showed that the higher the magnetic field strength the lower the stability, with the stability better between 100 and 500 Ω. However, above 500 Ω, the digesters showed poor instability, with 1000 Ω revealing the highest instability. Instead, the best performing digester was the 500 Ω as it stabilized at a higher magnetic field strength of 4.50 mT.

Power density is one of the most significant parameters of a microbial fuel cell. In a microbial fuel cell, like any other power source, the goal is to maximize output power [48]. The cell potential adjusts to the external resistance [49]. As the MFC is a source of voltage (V), where deviations taking place on the voltage can be regarded insignificant, in other words, lower frequency [50], the maximum power transfer theorem suggests that optimum power is generated when internal resistance (R_int_) and external resistance (R_ext_) of electricity source are equal to each other. It can be said that the power produced (P) is a function of current (I). The total resistance in the system is the sum of R_int_ and R_ext_. Therefore, according to Ohm’s law, current can be defined as:(1)$$I=\frac{V}{R_{\mathrm{int}}+R_{\mathrm{ext}}}$$

Therefore,
(2)$$P=I^{2}\times R_{\mathrm{ext}}={\left(\frac{V}{R_{\mathrm{int}}+R_{\mathrm{ext}}}\right)}^{2}\times R_{\mathrm{ext}}=\frac{V^{2}}{\frac{R_{\mathrm{int}}^{2}}{R_{\mathrm{ext}}}+2R_{\mathrm{int}}+R_{\mathrm{ext}}}\times R_{\mathrm{ext}}$$

The power condition is the utmost when the denominator is minimized. Therefore, the derivative of the denominator with respect to an external resistance,
(3)$$\frac{d}{{\mathrm{dR}}_{\mathrm{ext}}}\left(\frac{R_{\mathrm{int}}^{2}}{R_{\mathrm{ext}}}+2R_{\mathrm{int}}+R_{\mathrm{ext}}\right)=-\frac{R_{\mathrm{int}}^{2}}{R_{\mathrm{ext}}^{2}}+1$$

To obtain the maximum power condition, the first derivative should be equated to zero. Thus,
(4)$$-\frac{R_{\mathrm{int}}^{2}}{R_{\mathrm{ext}}^{2}}+1=0$$

This means that the power produced (P) is highest when $R_{\mathrm{int}}=R_{\mathrm{ext}}$.

When external resistance is either lower or higher than internal resistance, the power produced will decline. Figure 7 displays the effect of digester type on the power density. For all digesters, the power density firstly increased, then stabilized at maximum power density (i.e., where $R_{\mathrm{ext}}$ = $R_{\mathrm{int}}$) and lastly decreased. In our study, the highest power production of 30.17 mW/m^2^ was obtained at an external resistance of 500 Ω. The same conclusion was suggested by Lyon et al. [41], who found that when the internal resistance was 300 Ω, the highest generation of power was found at an external resistance of 470 Ω, after that 1000 Ω, 100 Ω, 10,000 Ω and lastly 10 Ω. The external resistance of 100 Ω generated the lowest power production possible since the external resistance was too low compared to the internal resistance [49]. It is evident from the graph that an increase in external resistance from 100 Ω to 500 Ω led to a maximum power density increase of 27.0 mW/m^2^. On the other hand, the external resistance of the 1000 Ω generated the highest power generation in the hydrolysis stage. However, after passing the hydrolysis stage, the power generation of this digester decreased drastically perhaps due to the very high external resistance in the system.

Figure 8 portrays the maximum power density as a function of external resistance or load resistor. It is evident from the graph that, for low resistance, an increase in external resistance was accompanied by an increase in maximum power density. However, after the optimum external resistance of 500 Ω, the increase in external resistance resulted in a decrease in power density. The same observation was found by Carreon-Bautista et al. [51], who recorded the maximum power density when varying the load resistance value between the electrodes and found that, external resistance was proportional to maximum power density for load resistance below 400 Ω and that external resistance was inversely proportional to maximum power density for load resistance above 400 Ω.

Coulombic efficiency is the effectiveness with which electrons are conveyed in a system to complete an electrochemical mechanism. This is a significant measure of the efficiency of MFC since it determines the total amount of coulombs retrieved as electrical current. Figure 9 shows coulombic efficiency and electrical conductivity concerning external resistance. For external resistance below 500 Ω, both coulombic efficiency and electrical conductivity increased with an increase in resistance, whereas above 500 Ω, external resistance hindered the system. As was the case above, the use of 500 Ω external resistance performed better as indicated by the high coulombic efficiency and electrical conductivity. The coulombic efficiency relies on the resistance or impedance of the system, and greater resistance results in a reduction in the coulombic efficiency because it is difficult to recover electrons from a process with too much resistance [52]. Therefore, this suggested that the 500 Ω digester had the least resistance; the high coulombic efficiency is an indication that the digester retrieved the highest number of electrons.

The effect of electrical conductivity on both coulombic efficiency and maximum power density is shown in Figure 10. It is evident from the graph that, an increase in electrical conductivity was accompanied by an increase in both coulombic efficiency and maximum power density. Therefore, it is the high electrical conductivity of the 500 Ω that made it to generate the highest coulombic efficiency of 81.2% and maximum power density of 30.17 mW/m^2^. The higher electrical conductivity (209.3 μS/cm) certainly enhanced the way this digester performed as it reduced the ohmic losses (resistance that ions and protons encounter when they move inside a MFC) in the digester; the higher ions and protons present in the solution accelerated the current flow in the external circuit [27], and ultimately, power density.

### 3.4. Process Stability Indicator

Even though the above results confirmed that external resistance affects electrochemical efficiencies as well as the performance of the digesters, it is worth knowing whether the behavior of protons and ions (as expressed in terms of electrochemical efficiencies) affected stability.

Stability may be described by the concepts of resilience (the system’s recovery rate after a disturbance) and resistance (the system’s ability to withstand disturbance) [53]. Therefore, the choice of external resistances and pH of the system play a vital role in the total efficiency of the microbial fuel cell [50]. Figure 11 depicts the influence of digester type on process stability as a measure of pH. For all digesters, the pH increased with an increase in days. The optimal pH for enhancing the electrochemically-active acetogens lies in the range 6.8–7.0, whereas for improving the performance of electrochemically-active methanogens, the optimum pH lies in the range of 7.0–7.5. The pH values of the 500 Ω digester were higher than the other digesters, with the highest pH of 8.6 obtained on day 30. However, the pH values from this investigation were within the above-mentioned optimal range.

Amongst the MFCs investigated, the 1000 Ω digester had the lowest pH values. The VFA (particularly acetic acid) is the chief contributor to methane generation. At very high levels, VFA is also the main inhibitor of anaerobic digestion. Therefore, the high biogas production and high current density of the 1000 Ω digester in the hydrolysis stage probably generated high VFA which led to inhibition of the system, as indicated by a fall in pH from 7.20 to 7.16 (day 1 to day 15).

The external resistance directly affects the availability of the anode as an electron receiver in the MFC. Moreover, external resistance exerts selective pressure on the exoelectrogenic microbial population. Because a lower external resistance (typically between 50 and 500 Ω) enhances microbial activity and growth, the transfer of an electron to the cathode electrode is enhanced [21,41]. Zhang et al. [54] suggested that external resistance in MFC controls the potential of the anode, allowing the exoelectrogens to balance the reduction electrode kinetics together with potential energy gain and therefore decreases activation losses. Also, the results of this study showed that raising the value of pH from 7.51 to 8.60 resulted in an increase in power density by almost 36 times (from 0.83 to 30.17 mW/m^2^). Our study (Table 2) also showed improvement in maximum power density by almost 36.4 times (from 0.83 to 30.17 mW/m^2^) when the pH was increased from 7.28 to 8.60. These outcomes showed that the interaction influence between external resistance and pH was significant, so the influence of pH on the generation of maximum power density was extremely dependent on the amount of external resistance.

The highest maximum power density of 30.17 mW/m^2^ was achieved at a pH of 8.60. Cordova-Bautista et al. [50] investigated the effect of external resistance and anodic pH on power density in MFC and found that the best-performing digester produced the highest power density of 405 mW/m^2^ at a pH of 8.60. On the other hand, a low power density of 0.83 mW/m^2^ and low pH value of 7.28 on the 1000 Ω digester indicated a loss of electrochemically active microbes probably due to the high build-up of protons and autotrophic microbes in the system.

### 3.5. Decontamination

Even though the pH of the digesters has been considered in the previous sub-section, it remains to be seen whether this stability indicator had a substantial influence on the pollutants of the digester. Figure 12 shows the effect of the digester type on total dissolved solids (TDS). For all digesters, the amount of total dissolved solids (TDS) decreased with time, which is evident in the degradation of biomass. However, each digester had a different rate of degradation.

Initially (before day 5), an increase in external resistance was accompanied by a decrease in TDS, and the digester with 1000 Ω had the lowest amount of TDS. This was probably due to the high magnetic field strength that was observed before day 5 (see Figure 6). The use of magnetic fields can influence the biological features as a result of the improvement of the activity of bacteria, which reduced the TDS of the 1000 Ω digester [55]. However, higher intensity and extensive retention time in a magnetic field can severely decrease the performance of a process, which is why the 1000 Ω digester did not reduce a significant amount of TDS after day 5 [47,56]. Instead, the digester with higher stability, i.e., 500 Ω digester, performed better as it revealed the lowest amount of TDS (0.2 ppm) present in the digester after day 30.

Figure 13 shows the effect of digester type on contaminants removed, namely TSS, TS, color, COD and VS. All MFC digesters had higher contaminant removals than the control, with over 60% removal. Among the contaminants analyzed, COD had the greatest treatability performance for external resistance between 300 Ω and 800 Ω. On the other hand, when the resistance was either very small (100 Ω) or very big (1000 Ω), color had the greatest treatability performance. The digester with 1000 Ω had the highest magnetic field strength of 7.12 mT. Our previous study [47] showed that exposure to higher magnetic field strength led to enhance color removal in wastewater. Other studies have also found the same observation [57,58].

The best-performing digester was the MFC with 500 Ω, which reduced contaminants by more than 89% on COD (98.6%), TS (97.4%), VS (96.1%), TSS (94.1%), and color (89.6%). This is mainly because of the higher stability (pH of 8.60) of the 500 Ω digester, which ultimately, resulted in better electron recovery efficiencies, viz. electrical conductivity (209.3 μS/cm), coulombic efficiency (81.2%) and maximum power density (23.3 mA/m^2^) [47].

### 3.6. Economic and Energetic Viability of the Systems

A techno-economic analysis was executed so as to determine the efficiency and economic viability of the microbial fuel cells. All energy-related estimations were based on methane yield. Biogas is chiefly made of methane (60%) and carbon dioxide (40%), and it was estimated that 80% of the energy produced by methane is converted into electrical energy [59,60]. The energetic viability of the systems was based on the biogas produced by the digesters: 100 Ω = 166.8 mL/day, 300 Ω = 282.2 mL/day, 500 Ω = 692.7 mL/day, 800 Ω = 473.0 mL/day, 1000 Ω = 115.5 mL/day, and control = 102.6 mL/day. Table 3 shows the cost calculation for all digesters.

The energy generated by methane per day (E_G_) is determined by Equation (5):(5)$$E_{G}={\;\dot{Q}}_{{\mathrm{CH}}_{4}}\times {\mathrm{LHV}}_{{\mathrm{CH}}_{4}}$$
where the term ${\;\dot{Q}}_{{\mathrm{CH}}_{4}}$ represents the rate of methane generated in m^3^ CH_4_/day, whereas the term ${\mathrm{LHV}}_{{\mathrm{CH}}_{4}}$ denotes the lower heating value, i.e., 35.8 kJ/m^3^ CH_4_.

Equation (6) can be used to calculate the energy needed by the water bath (E_B_):(6)$$E_{B}=\;\dot{m}\times C_{p}\times \Delta T=\;\dot{Q}\times \rho \times C_{p}\times \left(T_{1}-T_{0}\right)$$
where $\;\dot{m}$ (=$\;\;\dot{Q}\times \rho )$ is the mass flowrate in kg/day, $\;\dot{Q}$ denotes the substrate volumetric flowrate in m^3^/day, $C_{p}$ represents the specific heat capacity in kJ/kg.°C, ρ denotes the density in kg/m^3^, $T_{1}$ represents the temperature of the digester in °C, and $T_{0}$ is the substrate temperature in °C.

The energy generated by the MFC digesters (E_E_) can be obtained by Equation (7):(7)$$E_{E}=I\times V$$
where I represents the current in A, whereas V denotes the voltage in V.

The total energy generated by each digester was calculated as the difference between the energy generated by the system and the energy utilized by the system (Equation (8)):(8)$$E_{T}=E_{G}+E_{E}-E_{B}$$

With regards to the techno-economic analysis, the MFC digesters were more economical than the traditional anaerobic digester, i.e., control. The most economical digester was the MFC with an external resistance of 500 Ω, revealing an annual net energy profit of 48.22 ZAR/kWh (3.45 USD/kWh), which was about 19.3 times that of the traditional anaerobic digester (2.50 ZAR/kWh or 0.02 USD/kWh).

## 4. Conclusions and Recommendations

The results of this study suggested that external resistance is a factor that influenced biogas accumulation, electrochemical efficiencies, stability and decontamination. High resistance (1000 Ω) enhanced the performance of the system in the early stages of the digestion process. However, after passing the hydrolysis stage, high external resistance hindered the system, as it revealed low biogas accumulation, low electrochemical efficiencies and higher instability. On the other hand, moderate external resistance performed better, and the best-performing digester was the 500 Ω as it reached 692.7 mL biogas/g VS_fed_, which was approximately 6.8 times the 102.6 mL biogas/g VS_fed_ of the control. This was accompanied by high contaminant removals for COD (98.6%), TS (97.4%), TSS (94.1%), VS (96.1%) and color (89.6%). In terms of electrochemical efficiencies, higher coulombic efficiency (81.2%), electrical conductivity (209.3 μS/cm) and maximum current density (70.0 mA/m^2^) were also obtained from the digester with 500 Ω external resistance, which is an indication for the enrichment of electrochemically active bacteria. The digester also showed a higher maximum voltage generation of 0.431 V, which was about 12.7 times the 0.034 V of the digester with 100 Ω. It was observed that an increase in maximum power density was accompanied by an increase in pH, with the 500 Ω digester generating the greatest maximum power density of 30.17 mW/m^2^ at a pH of 8.60. The use of a MFC showed a net energy profit of over 3.38 ZAR/kWh (0.24 USD/kWh) compared with the traditional anaerobic digester, with the 500 Ω digester producing the greatest annual energy profit. Thus, the use of magnetite-nanoparticles and MFC with an external resistance of 500 Ω seems very promising for use in bioelectrochemical biogas production and wastewater treatment.

Even though the results obtained showed that the synergistic use of magnetite-nanoparticles and MFC with an external resistance of 500 Ω enhanced biogas production and wastewater treatability, the practicability of this method is still questionable. The number of publications on surveys related to the field application of MFC are rising progressively. The method is on its way from lab-scale to practical implementation. However, scaling-up applications of MFC are constrained by bacteria-electrode interactions, design concepts, electrochemical restrictions and multidisciplinary methods of environmental biotechnology and electrochemistry. Therefore, future works should look at transferring the proposed method at the highest scale. In so doing, the concerns of maintaining the proportionate rate of energy harvesting and voltage reversal will have to be addressed.

Future studies should also look at the use of automation/online monitoring systems on the proposed MFC system, as this may be helpful in controlling real-scale digesters. More specifically, an automatic anaerobic digestion process control system will enable rapid process stabilization with fewer operation and maintenance difficulties. The ultimate goal will be to allow anaerobic digestion processes to operate firmly at their optimum capacity.

## Figures and Tables

**Figure 1 microorganisms-11-00643-f001:**
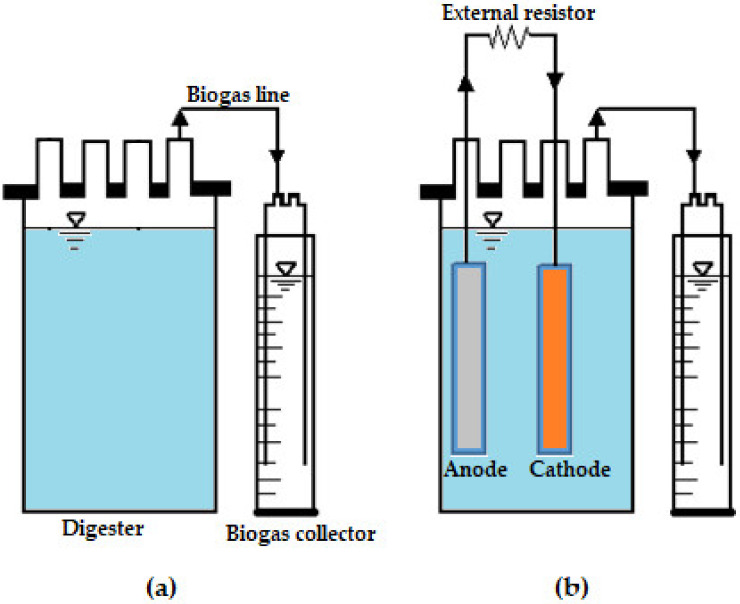
Schematic diagram of the digesters: (**a**) control with no external resistance; (**b**) and microbial fuel cell (MFC) with an external resistor.

**Figure 2 microorganisms-11-00643-f002:**
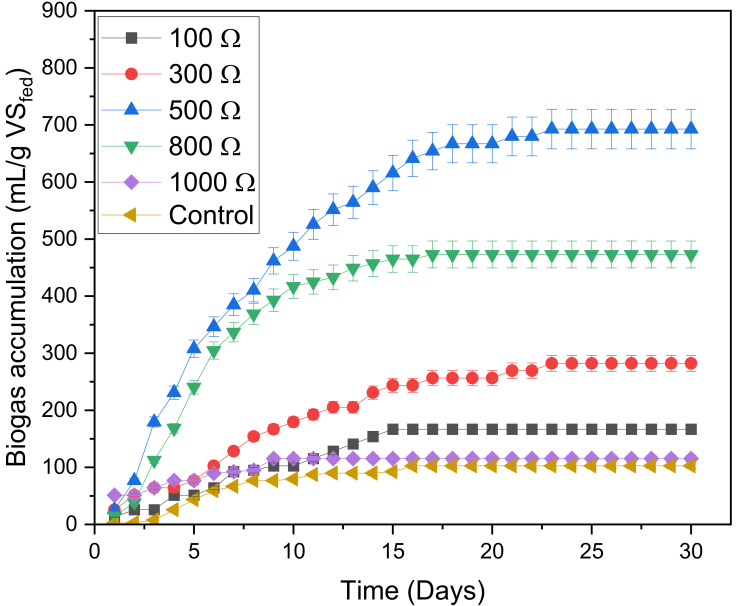
Biogas accumulation over a period of 30 days.

**Figure 3 microorganisms-11-00643-f003:**
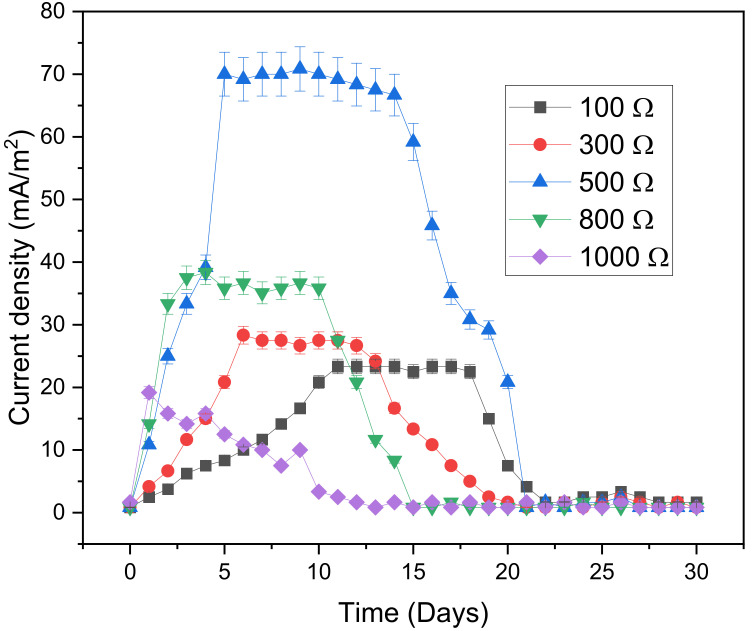
Variation of current density during the biofilm growth period.

**Figure 4 microorganisms-11-00643-f004:**
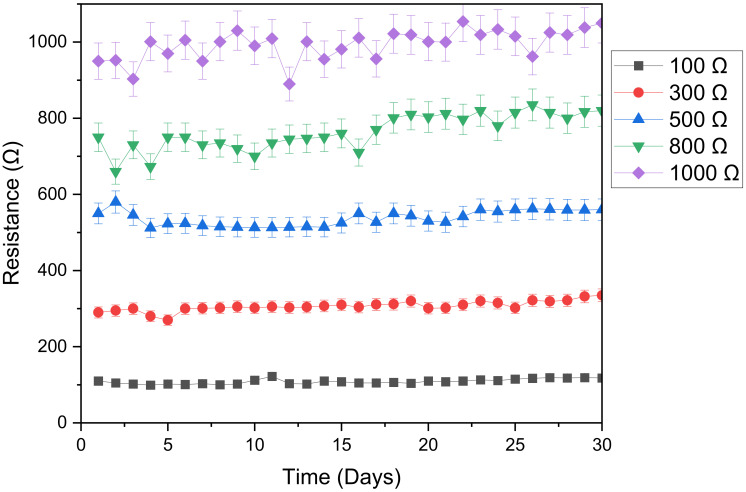
Effect of digester type on the overall resistance.

**Figure 5 microorganisms-11-00643-f005:**
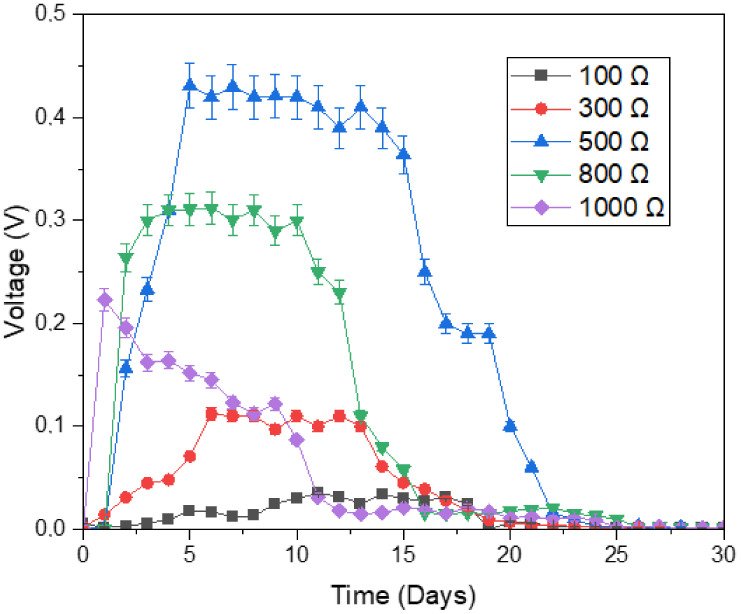
Effect of digester type on voltage generation.

**Figure 6 microorganisms-11-00643-f006:**
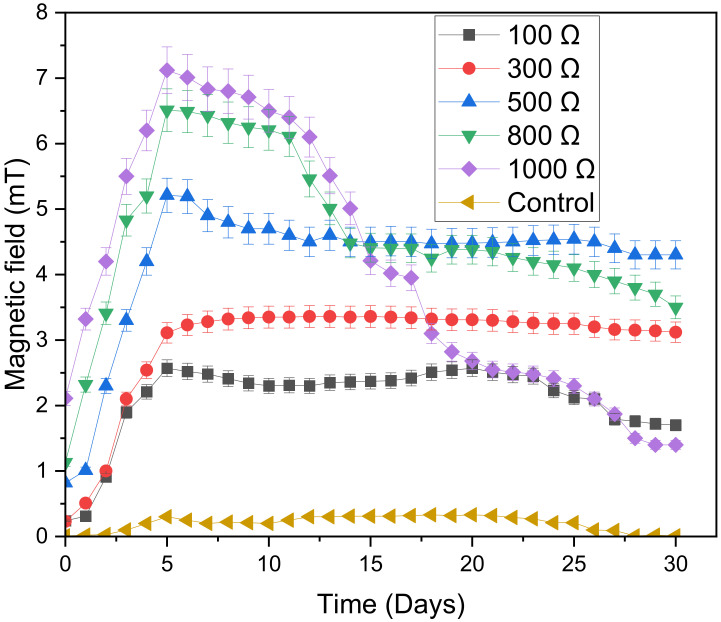
Effect of external resistance on magnetic field strength.

**Figure 7 microorganisms-11-00643-f007:**
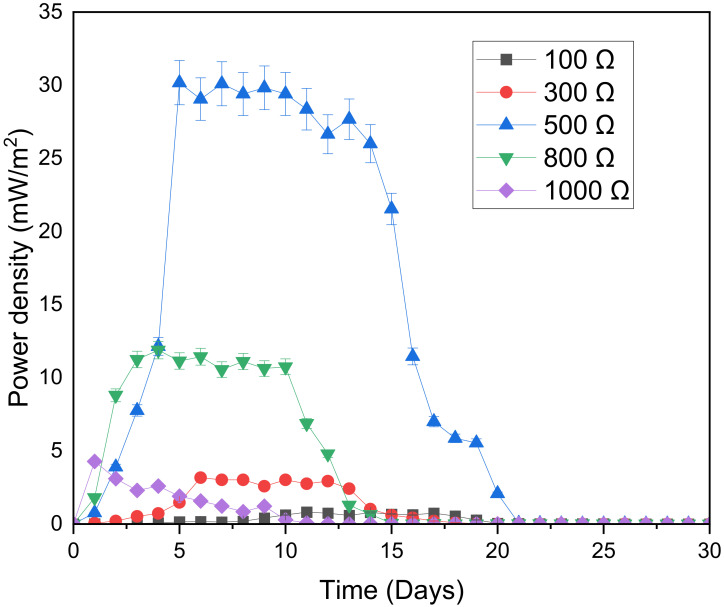
Effect of digester type on power density.

**Figure 8 microorganisms-11-00643-f008:**
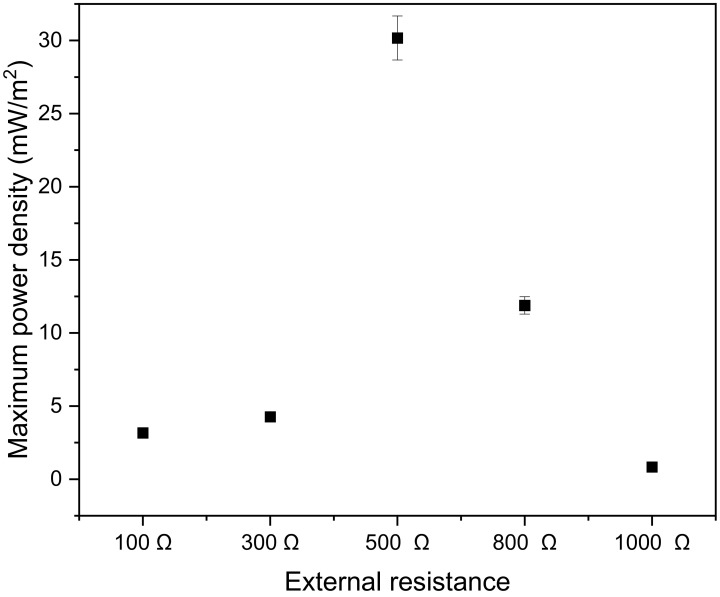
Power-resistance characteristics of the microbial fuel cell.

**Figure 9 microorganisms-11-00643-f009:**
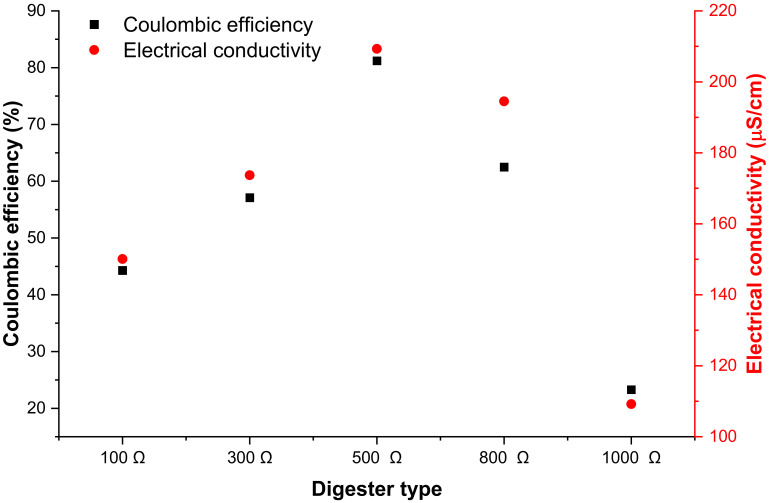
Coulombic efficiency and electrical conductivity with respect to external resistance.

**Figure 10 microorganisms-11-00643-f010:**
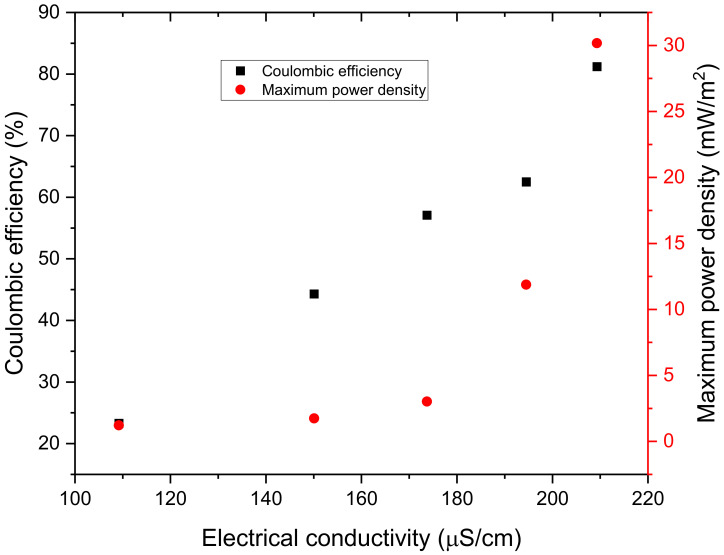
Coulombic efficiency and maximum power density with respect to electrical conductivity.

**Figure 11 microorganisms-11-00643-f011:**
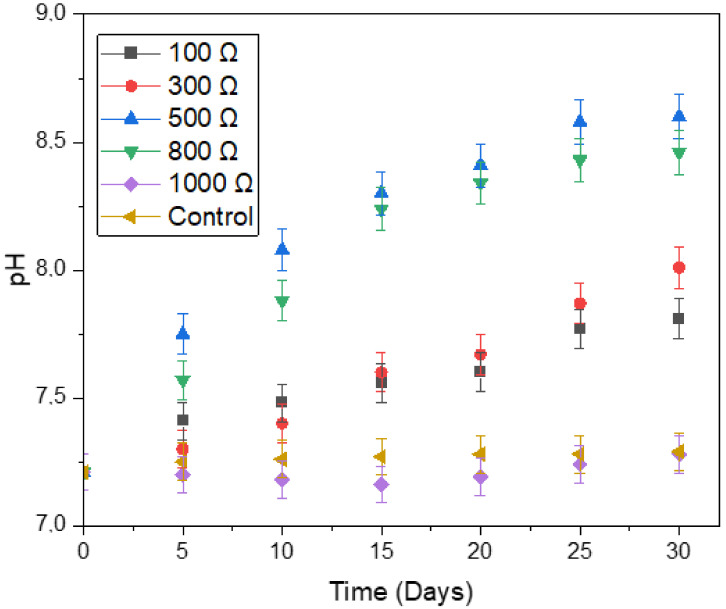
Effect of digester type on pH.

**Figure 12 microorganisms-11-00643-f012:**
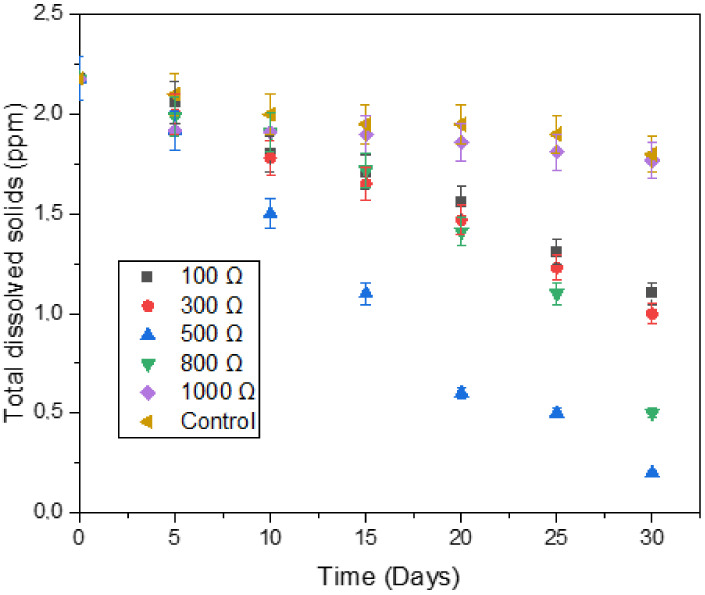
Effect of digester type on TDS.

**Figure 13 microorganisms-11-00643-f013:**
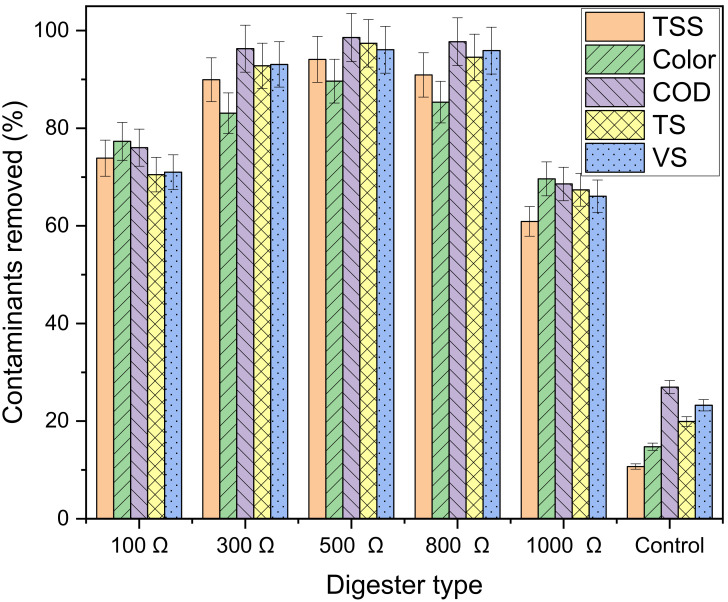
Effect of digester type on decontamination.

**Table 1 microorganisms-11-00643-t001:** Physical and chemical properties of the feed.

Parameter	Unit	Amount
pH	-	7.2 ± 0.3
VS	mg/L	45.6 ± 1.1
TS	mg/L	53.3 ± 5.3
Color	Pt.Co.	254.3 ± 6.3
COD	mg/L	2451.2 ± 200
TSS	mg/L	39.4 ± 1

**Table 2 microorganisms-11-00643-t002:** Influence of pH on maximum power density.

Digester Type	pH	Maximum Power Density (mW/m^2^)
1000 Ω	7.28 ± 0.27	0.83 ± 0.08
100 Ω	7.81 ± 0.15	3.17 ± 0.12
300 Ω	8.01 ± 0.14	4.27 ± 0.13
800 Ω	8.46 ± 0.10	11.88 ± 0.11
500 Ω	8.60 ± 0.11	30.17 ± 0.10

**Table 3 microorganisms-11-00643-t003:** Economic and energetic viability of the systems.

Type	Unit	100 Ω	300 Ω	500 Ω	800 Ω	1000 Ω	Control
Energy content of methane	m^3^/h	0.00149	0.00252	0.0062	0.0042	0.00253	0.00011
Energy generated by methane (E_G_)	kWh	0.00119	0.00202	0.0050	0.0034	0.00202	0.00035
Energy used by the water bath (E_B_)	kWh	0.00014	0.00014	0.00014	0.00014	0.00014	0.00014
Energy produced by the MFC (E_E_)	kW/h	0.00380	0.00512	0.03620	0.01426	0.00099	0.00000
Total energy (E_T_)	kWh	0.00486	0.00700	0.04102	0.01750	0.00288	0.00021
Daily energy profit (3.22 ZAR/kWh)	ZAR/kWh	0.01564	0.02255	0.13209	0.05636	0.00926	0.00068
Daily net energy profit (0.23 USD/kWh)	USD/kWh	0.00112	0.00161	0.00944	0.00403	0.00066	0.00005
Annual net energy profit	ZAR/kWh	5.71	8.23	48.22	20.57	3.38	2.50
Annual net energy profit	USD/kWh	0.41	0.59	3.45	1.46	0.24	0.02

## Data Availability

Not applicable.

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
