# Peer review of "Application of Magnetite-Nanoparticles and Microbial Fuel Cell on Anaerobic Digestion: Influence of External Resistance"

_microorganisms, 2023, doi:10.3390/microorganisms11030643_

Round 1

Reviewer 1 Report

The application of magnetite-nanoparticles and microbial fuel cell studied on the anaerobic digestion of sewage sludge. The digester with an external resistor of 500 Ω showed a high potential for use in bioelectrochemical biogas generation and contaminant removal for sewage sludge. The topic increases knowledge in the specific (and not general) field in the section of microbial biotechnology. It increases the knowledge in the field with complete experiments and controls. Conclusions are consistent with the presentation of data.

Manuscript with well written introduction, results of data are clear reported and the discussion of data seems complete, concise conclusions, updated bibliography and clear tables and figures.

Author Response

Thank you so much for your comments.

Reviewer 2 Report

Great read. I am impressed by how the results are presented. The only concerns I have are:

1) The authors need to touch upon economics for this novel AD system which can provide insights on how this technology compares to traditional AD systems.

2) I would also urge authors to include caveats of this study in the conclusions section to provide useful information on key hot spots that help improve the technology.

Author Response

Thank you so much for your comments. Kindly see attached response to your comments

Reviewer 3 Report

This paper, entitled Application of Magnetite-Nanoparticles and Microbial Fuel Cell on Anaerobic Digestion: Influence of External Resistance, is a scholarly work and can increase knowledge on this domain. The authors provide an interesting and original study, the content is relevant to Microorganisms. This study is in the spotlight of the current research work and can forsure increase knowledge on this domain, and generate new knowledge.

I have some general and specific comments:

- the abstract and keywords are meaningful

 - the manuscript is quite well written and well related to existing literature

- the authors should better introduce the objectives and the main goal of this study in the Introduction section

- please improve the quality of the figure, the resolution is low and the figures must be improved

- about the biogas collector, please discuss about the choice of such system, is there any possibility to implement automatic counter? The accuracy of such system could be discussed, due to the low sensitivity of such method. Moreover, the experiments were carried out during 30 days, is there any requirement of filling the bottle during the experiment? What about the risk of loss of information if the bottle is empty and not detected?

- please provide references about protocols applied for measurement and analysis, there's a reference APHA but maybe some of them could be added. Please mention just the references with citation into the text and insert these references in the bibliography section.

- About Table 1, please check the significance of number of digits for data

- About nanoparticles, how were prepared these nanoparticles? Is it from commercial provider or are they prepared before experiments? Are we sure that there's only nanoparticles and not microparticles? Is there any size distribution made on them? Is there any iron content measurement?

- Please provide error bars in graph (Figure 1). Please express the results in NmL per gram of VS (and not only in mL, please normalize the results according to Pressure 1 atm and Temperature 273.15 K).

- Please provide error bars in Figure 3, Figure 4, Figure 5, Figure 6, Figure 7, Figure 8.

- Please provide error bars in Figure 11, 12 and 13

- Please provide accuracy for data in Table 2.

- Please discuss about costs analysis iof such approach. Please discuss also about applicability and possibility of transfer.

- Please discuss about future works and perspectives. What about the feasability to transfer such approach at highest scale? What could be the trends od such method?

- Please discuss about limitations and constraints, please mention also advantages and strengthness of such approach.

As it, the paper is not fully acceptable for publication and requires some amendments and additional data. I recommend the following decision: RECONSIDER AFTER MAJOR REVISION.

Author Response

Dear reviewer. Thank you so much for your comments. Kindly find attached response to your comments.

Round 2

Reviewer 2 Report

The revisions are satisfactory.

Reviewer 3 Report

The authors provide a revised version of their manuscript taking into account all the comments and requests of amendments made in the previous version. The authors provide detailed and justified answers, I agree with all these answers. As it, the manuscript is now fully acceptable for publication. I recommend the following decision: ACCEPT IN PRESENT FORM.